# Insights into Biochemical Sources and Diffuse Reflectance Spectral Features for Colorectal Cancer Detection and Localization

**DOI:** 10.3390/cancers14225715

**Published:** 2022-11-21

**Authors:** Marcelo Saito Nogueira, Siddra Maryam, Michael Amissah, Andrew McGuire, Chloe Spillane, Shane Killeen, Stefan Andersson-Engels, Micheal O’Riordain

**Affiliations:** 1Tyndall National Institute, Lee Maltings, Dyke Parade, T12 R5CP Cork, Ireland; siddra.maryam@tyndall.ie (S.M.); michael.amissah@tyndall.ie (M.A.); stefan.andersson-engels@tyndall.ie (S.A.-E.); 2Department of Physics, University College Cork, College Road, T12 K8AF Cork, Ireland; 3Department of Surgery, Mercy University Hospital, T12 WE28 Cork, Ireland; andymcg963@gmail.com (A.M.); spillanechloe6@gmail.com (C.S.); shane.killeen@hse.ie (S.K.); micheal.oriordain@ucc.ie (M.O.)

**Keywords:** diffuse reflectance spectroscopy, colorectal cancer, optical spectroscopy, optical diagnostics, colonoscopy, surgical guidance, metabolomics, biomedical optics, biophotonics, multivariate analysis, chemometrics

## Abstract

**Simple Summary:**

Colorectal cancer (CRC) is the third most common and second most deadly type of cancer worldwide. The early detection and accurate characterization of colorectal cancer are associated with improved outcomes. With increasing emphasis on early cancer detection and the use of minimally invasive microsurgical techniques, the development of accurate diagnostic technologies to identify tumors and define their boundaries in real time becomes of paramount importance. The current research identifies potential cancer biomarkers and associated light-based instrument specifications to manufacture next-generation medical devices for CRC detection. These specifications have been chosen so that miniaturized instruments can be integrated into colonoscopes. Cancer biomarkers are listed to enable the use of complementary biochemical methods to analyze biological tissues. The impact of using next-generation colonoscopes in reducing cancer deaths can be assessed once medical devices are manufactured.

**Abstract:**

Colorectal cancer (CRC) is the third most common and second most deadly type of cancer worldwide. Early detection not only reduces mortality but also improves patient prognosis by allowing the use of minimally invasive techniques to remove cancer while avoiding major surgery. Expanding the use of microsurgical techniques requires accurate diagnosis and delineation of the tumor margins in order to allow complete excision of cancer. We have used diffuse reflectance spectroscopy (DRS) to identify the main optical CRC biomarkers and to optimize parameters for the integration of such technologies into medical devices. A total number of 2889 diffuse reflectance spectra were collected in *ex vivo* specimens from 47 patients. Short source-detector distance (SDD) and long-SDD fiber-optic probes were employed to measure tissue layers from 0.5 to 1 mm and from 0.5 to 1.9 mm deep, respectively. The most important biomolecules contributing to differentiating DRS between tissue types were oxy- and deoxy-hemoglobin (Hb and HbO_2_), followed by water and lipid. Accurate tissue classification and potential DRS device miniaturization using Hb, HbO_2_, lipid and water data were achieved particularly well within the wavelength ranges 350–590 nm and 600–1230 nm for the short-SDD probe, and 380–400 nm, 420–610 nm, and 650–950 nm for the long-SDD probe.

## 1. Introduction

Colorectal cancer (CRC) is the third most common type of cancer worldwide, representing 11.3% (1.85 million) of diagnosed cancer cases and resulting in 10.2% (0.88 million) of cancer-related deaths in 2020 [1,2]. The large mortality caused by CRC can be attributed to late-stage detection leading to poor patient prognosis. This, along with the significant morbidity and cost associated with standard surgery and associated adjuvant treatment modalities has led to the widespread adoption of screening methods to detect cancer at an early stage [3]. This is paralleled by a proliferation of minimally invasive methods such as Endoscopic Mucosal Resection (EMR), Endoscopic Submucosal Dissection (ESD), Transanal endoscopic microsurgery (TEM) and Transanal Minimally Invasive Surgery (TAMIS) to deal with these early lesions [4]. Currently, standard CRC screening tests such as the detection of blood in the stool have high false positive rates and accurate CRC detection is achieved by colonoscopy followed by relevant biopsies. However, screening and early diagnostic are limited by suboptimal compliance and lack of accessibility. With this in mind, global initiatives to develop early CRC detection methods have focused on CRC biomarkers and monitoring biochemical changes in tissues and biofluids. The high specificity of such biomarker detection methods is associated with their ability to detect low concentrations of specific molecules [5]. In tissue, structural changes are related to molecular concentrations in the millimolar range, whereas metabolic, immunologic and genetic features are associated with micromolar, nanomolar, and picomolar ranges, respectively [5]. Metabolic markers (e.g., enzymes, pO_2_, pH, and minerals) and immunologic markers (e.g., growth factors, hormones and cytokines) can be directly expressed in tissue at sufficient levels for real-time and non-invasive detection by optical techniques. Therefore, *in vivo* optical detection of tissue metabolic biomarkers is attractive to identify and localize precisely the tumor area, for example at colonoscopy, with no requirements for sample preparation for analysis of tissue sections, biofluids, and other samples used for cancer screening purposes. However, optical techniques can only be integrated into medical devices (e.g., colonoscopes) if technology allows for miniaturization and cost-effectiveness. One of the main cost-effective and miniaturizable techniques capable of probing tissue endogenous biomolecules is diffuse reflectance spectroscopy (DRS), which includes elastic scattering spectroscopy and near-infrared (NIR) spectroscopy for point measurements and can be translated to imaging by using hyperspectral imaging techniques.

DRS is an optical technique capable of tissue identification based on its biochemical composition and microstructure [6,7,8,9,10]. DRS works by sending light to the interrogated biological tissue and detecting the diffusively reflected light (i.e., the light that emerges from the tissue surface after being scattered inside it). Since the detected light traveled inside the tissue, it contains information about the tissue’s optical properties such as scattering and absorption [11,12,13,14,15,16,17,18]. Light scattering is dependent on the refractive index mismatches of tissue including sets of molecules (e.g., collagen fibers and fibrils), organelles (e.g., mitochondria), cell membranes, and inhomogeneity of the intracellular and extracellular environment [10,19,20,21]. Therefore, scattering is associated with the tissue microstructure. On the other hand, absorption is associated with the tissue biomolecular composition, as its absorption bands are dependent on the molecular energy levels (including electronic [10,22,23,24,25,26,27,28,29,30,31,32,33,34,35,36,37,38,39,40,41,42,43,44,45], vibrational [46,47,48,49,50,51,52,53,54,55,56] and rotational levels [57]). Tissue absorption is related to biological chromophores (or absorbing biomolecules), which were previously investigated in a number of clinical and preclinical studies [58]. These chromophores include β-carotene, bile, bilirubin, ceroid, collagen, deoxyhemoglobin (Hb), oxyhemoglobin (HbO_2_), methemoglobin (MetHb), water, lipid, and melanin [58]. The chromophores probed with DRS are typically present in biomolecule concentrations from micromolar to nanomolar [5]. These concentrations describe metabolic and immunologic features of contrast between normal and cancer tissues [5].

Previous DRS studies have investigated the average concentration of specific tissue biomolecules (i.e., chromophores) based on spectral fitting models of diffuse reflectance, diffuse transmittance and fluorescence [24,26,32,58,59,60,61,62,63,64,65,66,67]. These concentrations were analyzed in clinical breast cancer studies about tumor vascularity [68], blood oxygenation [69], quantitative chemical information of oxy- and deoxyhemoglobin (HbO_2_ and Hb, respectively), water, and lipids [70,71,72]. Bile, blood, water and lipid concentrations were also reported for clinical studies on liver cancer [61,73]. In addition, previous NIR spectroscopy studies on brain, muscle, mammary, lung and prostate cancers of rats and mice reported altered vasculature, oxygen dynamics and HbO_2,_ Hb, water, lipid, protein and DNA content in tumor tissues [74,75,76,77,78,79,80,81,82]. In terms of CRC detection, DRS studies reported that hemoglobin concentration and blood oxygenation can be used to differentiate between normal and tumor tissues [83,84,85,86,87]. Indices of statistical differences in biomolecule concentrations have been reported for application in colorectal cancer surgery [88]. In addition, DRS was used together with machine learning methods for direct tissue classification. These studies include those distinguishing tumor tissue from healthy surrounding tissues in the oral cavity (head and neck cancer) [89], breast [90,91], lung [92], and liver [93,94]. Recent studies have investigated the detection of colorectal cancer during surgery [88,95,96,97,98,99,100,101]. In terms of primary colorectal cancer, DRS and related optical spectroscopic techniques have mainly been used to guide colonoscopy by distinguishing the normal mucosa and malignant tissue inside the colon (luminal side). However, DRS studies analyzing multiple chromophores in an extended NIR wavelength range into the short wavelength infrared (SWIR) region are rare. To the best of our knowledge, this is the first CRC study optimizing wavelength ranges of reflectance spectra based on the combination between statistical tests and machine learning methods. In addition, the extended wavelength range between 350 and 1920 nm has been probed and our analysis objectively determined the biomolecules most important for tissue classification based on their depth-dependent absorption and scattering. As a final novel point of our study, our objective analysis does not need any homogeneous-medium assumptions such as those required by reflectance spectral fitting models to work.

In this study, we analyzed the spectral regions of best diagnostic potential and the biochemical sources of the classification between normal mucosa and tumor tissues based on the DRS spectral machine-learning features associated with chromophore absorption. The importance of these sources for such classification was determined by (1) assessing the amplitude of partial least-squares components (PLSCs) at relevant wavelength ranges of tissue scattering and absorption, (2) comparing the shape of the PLSC loading curves and chromophore absorption spectra, and (3) evaluating the classification performance (sensitivity, specificity, accuracy and area under the receiver operating characteristic curve) for wavelength ranges of statistically significant difference at a significance level of *p* = 0.001 of *t*-test after ensuring normality by using Anderson-Darling and Lilliefors tests. Our study evaluated the tissue classification in both superficial and deeper layers by using probes with short and long source-to-detector distances (SDD) of 630 µm and 2500 µm, respectively. Biomolecules were probed in a wide wavelength range from 350 nm to 1920 nm, which allowed for the collection of tissue structural and biomolecular signals at different depths. Our DRS spectral dataset contained 2889 spectra of freshly excised *ex vivo* tissues of 47 patients, which was used for robust and reliable analysis of the tissue biochemistry.

## 2. Materials and Methods

### 2.1. Diffuse Reflectance Spectroscopy (DRS) Instrumentation

The DRS equipment used in this study, illustrated in Figure 1, consisted of a broadband light source (HL-2000-HP, Ocean Optics, Edinburgh, United Kingdom) with emission ranging from 350 nm to 2400 nm, a quadrifurcated fiber optic probe with source-to-detector distance (SDD) of 630 µm (BF46LS01 1-to-4 Fan-Out Bundle, Thorlabs, Munich, Germany), a trifurcated fiber optic probe with SDD of 2500 µm (Fibertech Optica, Anjou, Canada), a visible/near-infrared (NIR) wavelength spectrometer (QE-Pro, Ocean Optics, Edinburgh, UK) and a NIR/SWIR spectrometer (NIR-Quest, Ocean Optics, Edinburgh, UK). The fiber optic probes were made of low-OH silica in order to allow better transmission at the SWIR range. These probes were used for both illumination and collection of the reflected light to be detected by the spectrometers. The visible/NIR spectrometer collected light in the wavelength range between 350 nm and 1140 nm, while the NIR/SWIR spectrometer detects light from 1090 nm to 1920 nm. The overlapping region was used to merge the spectra into one broadband spectrum from 350 nm to 1920 nm. Once reflected light was detected by the spectrometers, the intensity readings were preprocessed in order to obtain the tissue DRS spectra according to Section 2.5.

### 2.2. Probing Superficial or Deeper Tissue Layers

By using the 630 µm SDD probe (630 µm fiber center-to-center distance, Figure 1), our DRS system collected reflectance signals from 0.5 to 1 mm deep into tissue (between 450 and 1590 nm). In this case, the probe contained 600 µm core diameter fibers for both illumination and collection and will be referred to as a short-SDD probe throughout this article. In order to collect light from deeper tissue layers (between 0.5 and 1.9 mm deep between 450 and 1590 nm), we used a 2500 µm SDD probe (long-SDD probe) containing one source fiber in the center and 10 collection fibers surrounding it. Each 5 collection fibers were positioned linearly in the proximal end of the fiber to match the slit of the visible/NIR and NIR/SWIR spectrometers to optimize the light detection configuration. The source and collection fibers of the long-SDD probe had 600 µm and 200 µm core diameters, respectively.

In order to estimate the chromophores and depth interrogated by each probe, we used a spectral fitting algorithm to extract the optical properties from our DRS measurements. The fitting was based on a look-up table of Monte Carlo simulations of the light propagation into tissues and iteratively modifying chromophore concentrations and scattering properties. As a probe depth estimate for each wavelength, we used the depth of the maximum fluence value at the mean position between the source and detector. Then, the minimum and maximum probed depth were reported in this study with the purpose of illustrating the independence of the datasets acquired with each probe and its impact on the evaluation of the biochemical composition of superficial or deeper tissue layers.

### 2.3. Optical Data Collection

Our data collection started with the background and reference measurements. Reference measurements were taken by positioning each probe on a specialized holder able to keep a fixed distance between the fiber optic probes and our reflectance standard (FWS-99-01c, Avian Technologies LLC, New London, CT, USA). Since the holder was closed to avoid interference from ambient light, it was used to take both reference and background measurements. Probe contamination was avoided by covering our probes with transparent polyvinyl chloride (PVC) film during measurements. After each set of measurements, the plastic film was removed and the probes were cleaned with ethanol 70%. The same probes were used for every clinical measurement throughout this study. Every measurement was performed by positioning the probe as close to a perpendicular angle to the tissue surface as possible. Measurements of both probes were performed at similar locations within millimeters of each other. Briefly, we positioned the sample on a board with a coordinate system, which allowed us to come back to a similar position for both probes. the experimental procedure is described in detail in [11,12]. After completing the data collection, the data were safely stored for subsequent analysis. In this study, we collected a total of 1363 spectra for the short SDD probe (630 µm SDD) and 1526 for the long SDD probe (2500 µm SDD). Each spectrum is an average of a triplicate measurement in the same tissue location.

### 2.4. Clinical Protocol and Research Ethics

Our study included 47 patients undergoing bowel resection at the Mercy University Hospital (Cork, Ireland). Patient demographics and tumor characteristics are shown in Table 1. The study was approved by the Clinical Research Ethics Committee of the University College Cork. Our procedure consisted of collecting around 15 measurements of *ex vivo* mucosal tissues and 15 tumor tissues on the specimen after surgical resection. Measurements were taken from a typical area of 100 cm^2^. After a specimen was resected, the colonic lumen was exposed. The specimen was rinsed with water and cleaned afterward in order to remove the excess blood and any remaining feces from the mucosal surface. Then, the mucosa and tumor regions were identified by experienced surgeons. The time between the specimen removal and the start of the data collection was on average 40 min. All data collection was performed within an average time of 1 h after surgical resection. Physiological conditions were kept as much as possible throughout the data collection by keeping the tissue moist with a wet wipe. In order to correlate spectral readings with the tissues measured, the coordinate of every reading was registered by using a picture of the specimen over a grid. The boundary of each tissue type was determined by experienced surgeons. After the acquisition of all-optical DRS data, the specimen was returned to the Pathology Department for processing and analysis according to standard protocols. The ground truth of cancer tissue types was obtained by histopathology analysis.

### 2.5. Data Preprocessing and Feature Selection

First, both visible and NIR tissue intensity spectra had their background subtracted and the resulting signal was divided by the reflected intensity of the reference (reflectance standard) according to the expression:
(1)
$$Reflectance\left(\lambda \right)=\frac{1}{Reference\;reflectivity}\times \frac{Tissue\;reflected\;intensity-Background\;intensity}{Reference\;reflected\;intensity-Background\;intensity}$$


Next, the broadband reflectance spectra were obtained by merging the visible and NIR spectra based on the overlapping spectral region between the two spectrometers (from 1090 nm to 1140 nm). The merging was performed by interpolating the overlapping region of the two spectrometers and performing the following weighted sum:
(2)
$$Reflectance\left(\lambda \right)=\overset{100}{\underset{i=0}{\sum }}\frac{\left(\left(100-i\right)\times reflectance\;of\;VIS\;spectrometer+i\times reflectance\;of\;NIR\;spectrometer\right)}{100}$$


The result is a smooth reflectance curve where the reflectance measured by each spectrometer has a higher contribution at their respective wavelength regions of higher sensitivity. As preparation for classification tests based on k-nearest neighbors (which may be prone to overfitting when using numerous variables in comparison to the sample size), feature extraction was performed by using partial-least squares (PLS).

PLS is a supervised method of orthogonal transformation used for linear dimension reduction in a given dataset. PLS is used to create a new set of linearly independent variables (partial least squares components or PLSCs) which maximize the covariance between the predictors (reflectance values for each wavelength) and responses (tissue types) [112]. As PLSCs are weighted sums of the original variables (or predictors), the combination of weights of the first PLSCs shows the wavelength regions which are responsible for most of the discrimination between two classes (tissue types). More details about the calculation of PLSCs and the importance of predictors for better tissue discrimination can be accessed from the publications of Brereton et al. [113] and Gromski et al. [114], respectively. We also emphasize that our PLSCs are the same as the principal components (PCs) of PLS.

In this study, the bias due to the incomparable scales of observations (reflectance values) at specific variables (reflectance at particular wavelengths) was avoided by scaling/normalizing observations between −1 and +1 for each wavelength. By compensating for the difference in scale on reflectance values, this scaling ensures the feature extraction equally takes into account the contributions of each wavelength on the new set of variables based on the PLS maximization of the discrimination between the two tissue types. This contribution is translated into the weights (loadings) of each wavelength on the variables selected to develop a tissue classification model. Finally, data preprocessing and analysis were performed using home-made MATLAB routines (MathWorks Inc., Natick, MA, USA).

### 2.6. Extraction of Spectral Features

The spectral features for differentiation between normal mucosa and cancer tissues were extracted by using the first four PLSCs, which were selected to avoid overfitting by stopping to include new PLSCs when accuracy increments were lower than approximately 2%. These features were interpreted based on the amplitude and spectral shape of PLSC loadings. The amplitude was used to determine the contribution of specific wavelength ranges to tissue classification, which were associated with the main absorbing ranges of typical tissue biomolecules (or chromophores). The spectral shape was associated with characteristics of the chromophore absorption spectra. More details of the interpretation of the spectral features are described in Section 4.2.

In order to provide information about which wavelength ranges were related to most of the tissue biomarkers associated with cancer detection, delineation and potential carcinogenesis, we evaluated the tissue classification performance parameters (sensitivity, specificity, accuracy and area under the receiver operating characteristic curve; AUC) achieved by using wavelength ranges selected via statistically significant difference verified by a student *t*-test. First, normal distributions of DRS readings at each wavelength and tissue type were verified through Anderson-Darling and Lilliefors normality tests. Next, a student *t*-test was applied to the same distributions. Wavelength ranges leading to *p* < 0.001 were selected for building our tissue classification model. This model was built by applying PLS to the data on the relevant wavelength ranges and selecting the four highest-order PLSCs to be used with a weighted KNN classification algorithm. The weighted KNN algorithm used 10 neighbors and squared inverse distance between observations.

Based on the results of the KNN classification, its performance was evaluated by using two-fold cross-validation for 20 iterations. Each iteration consists of randomly dividing the dataset into training and test sets of equal size. The classification model was generated by using the training set and, then, applied for tissue classification on the test set. The process is repeated with the first test set used as the training set and vice versa. At the end of this process, the output was the mean of each classification performance parameter. The process is repeated 20 times. Then, the mean and standard deviation of the output of the 20 iterations were calculated. The reproducibility of these parameters was evaluated by the obtained standard deviations.

A flowchart summarizing the spectral analysis is shown in Figure 2.

## 3. Results

### 3.1. Tissue Classification Features Based on PLSC Amplitudes

Our first analysis for the selection of the features relevant to tissue classification was based on the PLSC loadings. The interpretation of the PLSC loadings is described in Section 4.2. Figure 3 illustrates that the PLSC1 loadings of the short-SDD probe exhibit the highest and second-highest absolute amplitudes in the range from 600 nm to 1350 nm and from 350 nm to 600 nm, respectively. A similar behavior is observed on the amplitudes of the PLSC1 of the long-SDD probe, whose first, second and third highest absolute amplitudes occur between 450 nm and 600 nm, between 650 nm and 1350 nm, and between 1350 nm and 1900 nm, respectively. These amplitudes suggest that most of the differentiation between mucosal and cancerous tissues may originate from the absorption and scattering processes below 1350 nm. Processes at the visible wavelength range 350–600 nm are predominantly related to blood absorption, and information regarding the near-infrared range 700–1350 nm could be associated with relatively lower absorption of oxyhemoglobin (HbO_2_) and deoxyhemoglobin (Hb), lipid and water at slightly deeper tissue layers.

On the PLSC2, PLSC3 and PLSC4 of the short-SDD probe, overall higher absolute amplitudes can be found for wavelengths below 700 nm and above 1350 nm, whereas the same components for the long-SDD probe have amplitude loadings more uniformly distributed over the full spectral window between 350 nm and 1900 nm. Particularly for PLSC4 of the long-SDD probe, the amplitude is higher for wavelengths above 1350 nm. While the high amplitudes below 700 nm for the short-SDD probe reinforce potentially relevant features of tumor detection at the UV-visible region (dominated by blood absorption), tissue classification using other wavelengths is unclear from only absolute PLS-loading amplitudes. With this in mind, we analyzed the shape of the PLSCs based on the wavelength regions and spectral shape of the chromophore absorption spectra, as shown in Section 3.3. On the other hand, this analysis is subjective and does not allow us to define specific wavelength ranges, as these would depend on subjective thresholds of the amplitude of PLSC loadings and the choice of how many PLSCs to be considered for thresholding. In addition, although PLSC1 is more important for tissue classification, the weight of each PLSC on the importance of wavelength ranges is unclear.

### 3.2. Wavelength Selection and Tissue Classification

Since analyzing the PLSC loadings may lead to subjective observations, we performed a more objective analysis by selecting wavelengths through statistically significant differences in the *t*-test (*p* < 0.001). In order to ensure that *t*-test could be applied to our dataset, Anderson-Darling and Lilliefors normality tests were applied to the data of each individual wavelength. Normal distributions were identified for all the wavelengths of both probes, except for those between 536 nm and 545 nm for the short-SDD probe. However, we confirmed statistically significant difference (*p* < 0.001) is obtained at those wavelengths upon application of the Wilcoxon rank-sum test (data not shown) and obtained no difference in tissue classification by including or excluding this wavelength range (Table 2).

After confirmation of normality for most of the wavelengths, we applied a *t*-test for each wavelength. We used the wavelength ranges where a statistically significant difference (*p* < 0.001) was obtained (Figure 4) to select bands relevant for tissue classification.

By using the wavelength ranges of Figure 4 as well as combinations of those ranges for UV-visible or NIR, tissue PLS-KNN classification models were built and compared with the model using all the wavelengths. The classification performance of each model can be found in Table 2 and Table 3.

Table 2 indicates that the spectral regions 600–1230 nm and 350–590 nm are the first and second most important for tumor detection by using the short-SDD probe, as accuracy is higher for these wavelengths. When tissue classification is performed only with wavelengths of the first spectral region, the achieved sensitivity (79.8 ± 0.9)% and specificity (84.4 ± 1.4)% are comparable to that obtained by using all the selected wavelength regions combined. This result suggests that the difference between mucosa and tumors is generated from the combination of absorption of Hb, HbO_2_, lipid and water as well as an optical scattering of tissue layers slightly below the tissue surface. In particular, Hb and HbO_2_ may play a significant role in classification across the superficial tissue layers, as the spectral regions leading to the highest discrimination cover all their absorption wavelengths.

In terms of the combination of wavelength ranges, the specificity achieved by combining the 350–590 nm and 600–1230 nm wavelength ranges was lower than using 600–1230 nm alone. This lower classification performance may be associated with higher absorption and scattering properties between 350 and 590 nm, which may lead to higher separation between the centers of the mucosa and tumor distributions recognized by PLS, while less contrast between the two distributions was achieved due to higher variation in reflectance values. On the other hand, higher classification performance over all parameters was obtained by combining the ranges 1530–1700 nm and 1730–1850 nm. Then, the information provided by Hb and HbO_2_ in both wavelength ranges may be redundant, whereas signals associated with water and lipid absorption may be complementary. The information from all the selected wavelength ranges is also complementary, as their combination leads to higher classification performance compared to all ranges tested in this study, including a 3.6% higher specificity than that obtained by using all wavelengths (350–1920 nm).

Table 3 shows that wavelength ranges below 950 nm led to higher sensitivity than those above 1200 nm for the long-SDD probe. This indicates that Hb and HbO_2_ are the chromophores that most contribute to accurate tumor detection in deeper tissue layers. Similarly to the short-SDD probe, the highest classification performance was achieved by using wavelengths on the optical window (in this case, 650–950 nm). Additionally, since the performance obtained for the long-SDD probe is higher than that obtained by using the short-SDD probe, signals of deep tissue layers may contain more relevant information for tissue discrimination, and probes could be designed in future studies to obtain information from relevant tissue depths. The importance of deeper tissue layer information is further reinforced by the higher classification performance achieved by using the ultraviolet (380–400 nm) and visible (420–610 nm) wavelengths alone compared to the best performance obtained by using the short-SDD probe.

Although the higher performance achieved by using the range 650–950 nm compared to shorter wavelengths may be attributed to the contribution of water and lipid absorption in the optical window, NIR wavelength ranges led to relatively low classification performance. In this case, the most informative NIR range was 1250–1380 nm, where features of higher variations in lipid and water absorption can be simultaneously observed (Figure 5).

Similar behavior was observed between the classification performance achieved by using the long-SDD probe and the short-SDD probe. By combining wavelength ranges containing ultraviolet and visible wavelengths, lower classification performance was obtained compared to the 650–950 nm range alone. In addition, the combination of NIR wavelength ranges improved tissue classification. Those results suggest that information from water and lipid absorption at deeper tissue layers probed by using long SDDs may be complementary and useful for cancer detection, while signals from Hb and HbO_2_ absorption may not bring useful information for tissue classification. Finally, the combination of selected wavelength ranges led to the achievement of as good performance as using the entire wavelength range (350–1920 nm) investigated in this study.

### 3.3. Relationship between Tissue Classification Features and Tissue Biochemistry/Microstructure

Once the importance of wavelength bands and their combinations were assessed, we used the shape of the PLSC loadings to understand the biochemical and microstructural sources of tissue classification. This section covers the results of our analysis, whereas the interpretation of spectral features of tissue classification is discussed in Section 4.2. Our analysis comprised of the determination of contributions of tissue scattering and chromophore absorption based on the shape of the PLSC loadings at wavelength bands of highest absorption of tissue chromophores as well as the “flatness” of PLSC loadings (indicative of the predominant contribution of tissue scattering). The selection of wavelength bands for subsequent interpretation is illustrated in Figure 5.

Figure 5 shows the spectral ranges where particular features of specific chromophore absorption can be observed. As an example, the red region shows the features of oxyhemoglobin (HbO_2_) and deoxyhemoglobin (Hb), including bands between 380 nm and 450 nm (peaks of HbO_2_ at 414 nm, 542 nm and 576 nm [21], and peaks of Hb at 433 nm, 556 nm and 757 nm). In the case of the latter peak of Hb (at 757 nm), we assume the feature is from Hb instead of lipid (peak at 761 nm) due to the higher Hb absorption and potentially higher Hb concentration in biological tissues (resulting in overall higher tissue absorption due to Hb). In addition, the used bile spectrum (from Nachabe et al. [61]) includes the water contained in the sample, as is obvious from the spectral range above 950 nm When the shape of the PLSC loadings is relatively flat and monotonic as a function of wavelength, we attributed scattering as the most important factor contributing to tissue classification using such PLSC. It is important to remember that, even though the chromophore spectra are not shown for wavelengths longer than 1600 nm in Figure 5, double absorption peaks between 1700 nm and 1800 nm are associated with lipid, whereas the increase in absorption close to 1900 nm is related to water [115,116]. By analyzing the spectral shape of the PLSCs, we obtained Table 4 below.

Based on the loadings of PLCS1, Table 4 suggests that the features contributing to tissue classification between mucosa and tumor are mostly related to Hb, HbO_2_ and water for the short-SDD probe and the Hb, HbO_2_, water and lipid for the long-SDD probe. For both probes, absorption features from the same chromophores of PLSC1 appear in PLSC2, which also contain features of scattering in visible and near-infrared wavelength ranges. In addition, absorption features of Hb, HbO_2_, water and lipid can be observed on the loadings of PLSC3 and PLSC4 of both probes. On the other hand, features of lipid absorption appear only at the loadings of PLSC3 and PLSC4 of the short-SDD probe, whereas characteristics of the same chromophore are exhibited in the loadings of PLSC1, PLSC2, PLSC3, and PLSC4 of the long-SDD probe.

## 4. Discussion

### 4.1. Impact of Depth-Resolved Determination of Wavelength Ranges and Biomarkers for Tissue Classification

Our study investigated the most important wavelength regions for discriminating colorectal mucosa and cancer tissues and how these regions can be related to the biomarkers contributing to this discrimination (discussed in Section 4.2). Furthermore, we provide information for two tissue probed depths, as these depths are dependent on the source-to-detector distance of our fiber optic probes (Section 4.2). Our optimization of wavelength ranges of reflectance spectra and evaluation of main biomarkers contributing to the classification of normal mucosa and tumor tissues is an extension of our previous work [11] showing the successful classification of these tissues by using support vector machines, as well as our study [12] estimating biomolecule concentrations by using a spectral fitting algorithm based on Monte Carlo simulations of light propagation in tissues (assuming homogeneous tissue models).

To the best of our knowledge, previous DRS studies did not perform an objective analysis of the importance of wavelength ranges for discrimination between mucosa and cancer tissues. These studies have been reviewed in our previous publication [11] and have only investigated specific wavelength ranges from 300 to 800 nm or from 900 to 2500 nm. No comparison between wavelength ranges objectively selected by statistical methods has been performed by using DRS for CRC detection. With that in mind, our study provides the objective analysis of the importance of ultraviolet, visible and near-infrared wavelength ranges for classification between colorectal mucosa and tumor tissues, which is especially useful (1) to design new optical spectroscopy and imaging systems restricted to specific wavelength ranges (for cost-effectiveness, performance maintenance upon miniaturization and integration into medical devices, higher accuracy at specific wavelengths, and higher spatial resolution), (2) to determine the range of probed tissue depths where improved tissue classification can be achieved and where optical tissue biomarkers tend to significantly influence tissue classification, (3) to identify which tissue biomarkers are the most important for discrimination between mucosa and cancer tissues based on DRS signals.

Our study improves the subjectivity of analysis of amplitude and shape of PLSC loadings (Figure 3) by adding an objective analysis based on the selection wavelength ranges for tissue classification based on *p* < 0.001 (statistically significant difference) for the *t*-test and comparison of PLS-KNN classification performance metrics by using each combination of selected wavelength ranges. Higher classification performance at wavelength ranges of specific tissue chromophores (biomolecules) indicates the most important wavelength ranges and respective biomarkers to classify normal mucosa and cancer (Section 3.2). This classification performance adds to biomarker identification based on features of tissue scattering and chromophore absorption based on the amplitude (Section 3.1) and shape (Section 3.3) of PLSC loadings since statistical wavelength selection and subsequent classification cannot identify scattering contributions spread out all wavelengths. The scattering contribution to tissue classification is only observed by analysis of PLSC loading shapes as a function of wavelength.

In this study, the analysis of PLSC loadings suggests that tissue scattering is secondary but still important for tissue classification since only PLSC2 loadings resemble scattering coefficient curves as a function of wavelength (Section 3.3). However, it is worth noting that different combinations of optical properties (scattering and absorption coefficients) lead to probing a different depth in tissue. Probing different depths means that each wavelength of each DRS spectrum extracts biomarkers (chromophore concentrations and scattering properties) at a different tissue depth. Additionally, the higher source-detector distance increases the chances of collecting light which traveled longer and deeper into tissue. Therefore, short-SDD and long-SDD probes capture information on biomarkers at different depths, as these depths depend on both probe geometry and tissue optical properties. Based on Monte Carlo simulations of light propagation in tissues performed in our previous study [11], the probed depth of the short-SDD probe was mostly within 0.5–1 mm for wavelengths between 450 and 1590 nm, whereas that of the long-SDD probe varied between 0.5 and 1.9 mm for the same wavelength range. Statistical methods presented in this paper enable objective depth-resolved biomarker identification and selection of wavelength ranges. This depth-resolved analysis is not achievable by spectral fitting models assuming that tissue is homogeneous because tissue heterogeneity is neglected to keep the number of fitted parameters to a minimum. At the cost of a tissue homogeneity assumption, such spectral fitting models can retrieve average tissue scattering properties and chromophore concentrations. Although average concentrations are easy to interpret, they contain neither information about tissue depths nor wavelength ranges to best differentiate normal and cancerous tissues.

Previous studies have calculated sensitivity and specificity for tumor detection based on biomolecular concentrations obtained from assumptions of homogeneous media, especially using spectral fitting models of diffuse reflectance, transmittance and fluorescence [24,26,32,58,59,60,61,62,63,64,65,66,67]. In imaging and tomography applications, these concentrations are typically extracted from the absorption at a few wavelengths and used as diagnostic indexes in different applications [117], whereas point spectroscopy evaluates a larger number and often a wider range of wavelengths. However, the potential of point spectroscopy is not fully exploited if homogeneous media assumptions are made, as useful information can potentially be extracted from such a larger number of wavelengths and depths probed. Our previous work [12] has shown that biomolecular concentrations are different depending on the probed depth by varying the probe SDD and by using a spectral fitting model for homogeneous tissue. In this study, the importance of measurements at each wavelength is considered separately. The probed depth of reflectance at each wavelength is different and incorporated into our analysis using statistical tests and machine learning methods. Analyzing DRS measurements of each wavelength separately means that information is extracted from tissue biochemistry and microstructure at multiple depths. Therefore, the importance of chromophores obtained in this study is based on more complete information compared with previous studies and exploits the full potential of DRS point spectroscopy measurements. By using our depth-resolved analysis, tissue biomarker information was interpreted by using PLSC loadings while the objective selection of optimized tissue classification parameters was determined by evaluating classification performance directly.

### 4.2. Spectral Features for Colorectal Cancer (CRC) Detection

The present study investigated the spectral regions and biomolecules associated with cancer development by identifying the most important spectral features for the differentiation between normal and cancerous tissues. This identification was performed by using partial least-squares (PLS). One of the advantages of using PLS methods is that they can provide insight into the variables most likely to be responsible for the differentiation between two groups via the interpretation of weights and loadings. With this in mind, PLS is typically used in exploratory studies focusing on which variables are best discriminators [113]. In molecular biology applications (e.g., metabolomics, proteomics, lipidomics, glycomics and others), these variables can be related to biomolecules via features of their generated physical processes present in the measured signal (e.g., the fluorescence emission of specific molecules in a tissue fluorescence measurement) [114]. In terms of optical techniques, this approach was used by Wang et. al., who found that NADH, collagen, and porphyrin were related to oral cancer detection by using fluorescence spectroscopy [118]. In this study, biomolecular contribution to the DRS signal is related to absorption features of tissue chromophores such as oxyhemoglobin (HbO_2_) and deoxyhemoglobin (Hb), lipid and water, as well as scattering tissue features related to size and refractive index of the main scatterers. As a result, absorption and scattering features related to discrimination can be observed on the PLS loadings, since these loadings are based on the differentiation between mucosa and tumor tissues present on the DRS signal.

PLSC loadings can be interpreted based on their amplitude and spectral shape. Higher absolute amplitudes of PLSC loadings on spectral regions of absorption of specific molecules mean these molecules are associated with the PLSC(s). The same association is reinforced if the spectral shape of the PLSC loadings is similar to that of the biomolecular absorption spectra within the pertinent spectral regions. The first PLSCs are more relevant for tissue discrimination. Therefore, analyzing the absolute amplitudes and spectral shape of loadings of the first PLSCs allows us to determine which biomolecules are potentially more relevant for differentiating mucosal and cancerous tissues.

By analyzing parameters such as loadings of principal component analysis (PCA) or PLS, it is possible to determine which biomolecules are involved with the classification between groups of tissues. The loadings of PLS components are typically analyzed in metabolomics [114], where the focus is determining which variables (related to chemicals) are best discriminators for certain tissue groups (e.g., cancerous and healthy tissue) rather than understanding whether tissue classification is possible [113]. A similar rationale is used to analyze spectral features associated with other physical phenomena related to biomolecules. For optical techniques such as fluorescence spectroscopy, biomolecules include NADH, FAD, collagen, elastin, porphyrins and lipopigments [24,26,27,32,60,67,119,120,121], whereas Fourier-transform infrared (FTIR) spectroscopy and Raman spectroscopy probe chemicals such as amino acids, proteins, lipids, carbohydrates, nucleic acids, porphyrins, and water [47,49,122,123]. A more extensive list of altered biochemical composition in CRC was reported by previous studies on lipidomics, proteomics, metabolomics, genomics, glycomics and other molecular biology sciences [124,125,126,127,128,129,130,131,132,133]. Out of all biomolecules in the extensive list of CRC tissue discriminators, our DRS study focuses on biomolecules with detectable absorption (chromophores) in the wavelength range between 350 and 1920 nm. These biomolecules include β-carotene, bile, bilirubin, ceroid, collagen, deoxyhemoglobin (Hb), oxyhemoglobin (HbO_2_), methemoglobin (MetHb), water, lipid, and melanin.

Previous studies using PLS to investigate biomolecular features in optical measurements are scarce. By using fluorescence spectroscopy, Wang et al. [118] used PLS loadings to determine that the biomolecules involved with oral cancer detection were collagen, NADH and porphyrins. This determination was based on the peaks of the loadings as a function of wavelength, which occurred at 390 nm and 470 nm for the 320 nm excitation, and at 460 nm and 640 nm for the 360 nm excitation. Apart from the study of Wang et. al., our study used the amplitudes and shapes of PLSC loadings to identify CRC biomarkers and corresponding wavelength ranges based on the loadings of the four first PLSCs. These loadings suggest that only Hb, HbO_2_, lipid, water and light scattering in tissue contribute to PLS features to be used for tissue classification (Section 3.3). In a qualitative analysis based on the amplitude of PLSC1 loadings, Hb and HbO_2_ were the most important biomolecules for tissue classification since the highest and second highest absolute amplitudes (Figure 3) occurred between 350 and 1350 nm for both probes used (Section 3.1). Light scattering, lipid and water were of secondary importance, since only PLSC2 appears to contribute to scattering, and the wavelength range between 1350 and 1900 nm has the third highest absolute amplitude only for PLSC1 of the long-SDD probe (Section 3.1). It is important to note that all PLSCs used for tissue classification exhibit spectral features of water, and all PLSCs except PLSC1 and PLSC2 of the short-SDD probe have lipid features. The absence of PLSC1 and PLSC2 features for lipids suggests that no superficial lipid signal up to 1 mm deep contributes to CRC detection, as the maximum probed depth for the short-SDD probe is ~1 mm. Additionally, scattering may have contributed to PLSCs other than PLSC2 even though its contribution may not show any strong trend on the PLSC loadings (Figure 3). Finally, one reason the scattering properties may not be more relevant for tissue classification is due to the relatively short source-detector distance providing a low sensitivity to alterations in scattering properties between tissue types. This means there might be a scattering-related contrast between tissues that our instrument is not sensitive to because of our future endeavors of translating our findings to an endoscopic *in vivo* study requiring that the short-SDD probe and all supporting equipment fit within the endoscope. Non-endoscopic applications targeting the classification of normal mucosa and cancer tissues relying on longer SDD and/or quantities associated with tissue scattering may still be useful to find scattering differences between these tissues.

### 4.3. Considerations on Biomolecular Concentrations and Probed Depth for CRC Detection

Based on the current literature evidence that Hb, HbO_2_, lipid and water concentrations differ in normal tissues and tumors, our results agree that these biomolecules can be used for tumor detection. In our study, the features of each biomolecule were illustrated in the shape of the PLSC loadings, while the molecular relevance for differentiation between colorectal mucosa and tumor could be determined with the amplitude of the loadings and higher classification performance of selected wavelength ranges. In general, there is an agreement between the wavelength ranges leading to higher classification performance and higher absolute amplitudes of the PLSC1 loadings. Since taking these ranges solely based on the loadings is subjective, we used a more objective analysis by selecting ranges of *p* < 0.001 in the *t*-test. This analysis suggested that Hb and HbO_2_ information relevant to tumor detection can be collected from specific wavelength ranges within the UV-visible region and the optical window. A combination of these ranges may not lead to more accurate tissue classification. On the other hand, classification can be improved by probing water and lipid at several near-infrared (NIR) ranges which provide complementary information for tissue discrimination. Our results regarding the importance of Hb, HbO_2_, lipid, water and scattering for CRC detection agree with results based on our objective statistical analysis and add to the results of our previous work [11], which suggested that wavelengths in the optical window contribute to higher accuracies of CRC detection. When neglecting local blood oxygen saturation (StO_2_) for reflectance spectral fitting under the homogeneous medium assumption, it is worth noting that lipid and scattering contributions to tissue classification become more important than contributions of total hemoglobin content [12]. However, when considering StO_2_ and distribution of chromophores scattering properties over all probed tissue layers in DRS, near-infrared spectroscopy, elastic scattering spectroscopy and hyperspectral imaging, previous *in vivo* and *ex vivo* studies have shown that contributions of Hb and HbO_2_ are comparable to those of lipid and water [11]. These contributions are evidenced by similar accuracies obtained when using wavelength ranges between 400 and 1000 nm (91.2 ± 0.9 accuracy) and between 1000 and 1920 nm (92.2 ± 1.3 accuracy) for the short-SDD probe [11].

In terms of wavelength ranges for differentiation between normal and cancer tissues, those in the optical window resulted in the highest classification performance for both short-SDD and long-SDD probes. This performance could have been achieved by considering the information on a number of biomolecules at variable depths and targeting wavelengths of higher light penetration. Our study suggests the latter is one of the main discriminatory factors, as probing deeper tissue layers with the long-SDD probe also led to higher performance compared with the short-SDD probe. Since more accurate tissue classification is achieved by using wavelengths at the optical window and using the long-SDD probe, the most relevant biomolecular changes associated with CRC detection, and potential carcinogenesis may occur at deeper tissue layers, which can be exploited by future studies.

Similar to the short-SDD probe, the highest classification performance was achieved by using wavelengths on the optical window (in this case, 650–950 nm). Additionally, since the performance obtained for the long-SDD probe is higher than that obtained by using the short-SDD probe, signals of deep tissue layers may contain more relevant information for tissue discrimination. This evidence can be further reinforced by the higher performance achieved by using the ultraviolet (380–400 nm) and visible (420–610 nm) wavelengths alone compared to the best performance obtained by using the short-SDD probe.

Our analysis of wavelength selection indicated that probing specific wavelength ranges for Hb, HbO_2_, water and lipid absorption lead to similar classification accuracies as those achieved by using the wavelength range 350–1920 nm (Table 2 and Table 3). These ranges may be used for future instrument design by keeping the geometrical configuration of the probe used in this study. In addition, extending the wavelength range towards longer NIR wavelengths may add information from other biomolecules as well as the contribution of the complementary data of water and lipid. With this in mind, future studies may investigate the complementarity of water and lipid information at wavelengths longer than 1920 nm and include features of other biomolecules absorbing at NIR and mid-infrared wavelengths (e.g., proteins, carbohydrates and nucleic acids).

### 4.4. Strength of the Cross-Validation of Our Model

In this study, we have used our PLS model to show the importance of objectively chosen wavelength ranges in tissue classification. Our tissue classification model was not used to assess the maximum classification performance to be obtained by testing several machine learning algorithms with our dataset. In fact, we have previously shown that higher classification performance can be obtained [11]. For correct interpretation of the influence of each wavelength range in the tissue classification, it is extremely important that our PLS model is general enough so that high classification performance parameters (sensitivity, specificity, accuracy and AUC) are not obtained due to overfitting by using large subsets of the dataset to build our classification model.

Although 5-fold cross-validation is frequently used for validation, this study used two-fold cross-validation in order to show a robust tissue classification in our dataset by using 50% of the data as a training set (two-fold) instead of 80% (5-fold). Two-fold cross-validation also allows the model to be tested in a larger dataset compared to 5-fold cross-validation, while the model is trained in a smaller subset of the total dataset. If the model is not general enough, training the model with smaller random subsets of the dataset while testing it in larger random subsets can lead to lower classification performance parameters. Therefore, robust classification through two-fold cross-validation means a stronger potential of generalization of the model upon an increase in sample size, especially compared to validation using more than 50% of the dataset for training and less than 50% for testing.

### 4.5. Limitations of Our Study

In order to evaluate oxygenation changes that could affect the results of the present study, we conducted a pilot observation of the DRS signal in 3 patients. We observed no significant variations in average Hb and HbO_2_ of 7 mucosa sites and 7 tumor sites 15 min from the beginning of our measurements (data not shown). In this case, the DRS signal was monitored every 5 min during the collection time period. In terms of expected biochemical differences when translating our findings to *in vivo* studies, we expect a similar behavior showed by Baltussen et al. [88] in fat, tumor, and healthy colorectal wall tissues. The authors exhibited a statistically significant difference in blood concentration (%) and StO_2_ (%) when measured *ex vivo* (within 1 h after resection) compared to in vivo. According to their study, both blood content and StO_2_ increased, presumably due to an increase in blood volume in the capillaries after excision [134], exposure to air and decreased oxygen consumption by the cells in the specimen. Their study also suggests the water content decreases due to dehydration (vaporization and leakage) of the resected tissue, which was minimized in this study by keeping the tissue moist with wet wipes. However, it is important to remember that measurements by Baltussen et al. were taken by probing a different tissue depth compared to our study (probe with SDD 1.29 mm center-to–center distance) and the analyzed tissues are different from colorectal mucosa. Additionally, the behavior of StO_2_ (%) is still not well understood, as results of Baltussen et al. disagrees with those of Salomatina et al. [135], who evaluated mouse ear tissues 5 to 10 min after excision (*ex vivo*) of tissue and after 24 and 72 h of storage.

Parts of our methodology are dependent on results achieved by using our dataset of almost 3000 measurements on 47 patients, which is assumed to be sufficiently robust for all analyses presented in this study. To ensure the wavelength selection was robust, the statistical test was based on all collected data. One should be aware that the wavelength selection was independent of the machine learning model. Therefore, classification performance metrics were affected by which wavelength ranges were selected, but not by the process to select wavelengths. It is worth noting that wavelengths were not used directly as descriptors/features of our machine learning (ML) models/classifiers. The four selected PLS components (PLSC) have been used as spectral features for tissue classification. A number of PLSCs have been selected as the minimum number of PLSCs before only increments of ~2% accuracy were gained upon inclusion of a new PLSC (data not shown). This threshold has been chosen to provide the best possible description of the data while avoiding overfitting. The four PLSCs were calculated on the entire dataset before two-fold cross-validation was performed for KNN models. Since PLSC loadings for tissue classification have been determined based on all the collected data (including the test set for the 20 iterations of two-fold cross-validation), it is important to note that the classification performance metrics were not calculated for optimized ML models in which spectral bands and PLSCs are selected at specific subsets of our dataset (discovery set) and subsequently training classifiers. Our classification performance metrics were used for the comparison of results obtained for each objectively/statistically selected wavelength range with the aim of placing wavelength ranges in the order of importance for tissue classification, and subsequently, the main cancer biomarkers upon the association between these ranges and tissue chromophores and scattering properties were identified. For the estimation of classification performance metrics obtained with wider wavelength ranges, one should check our previous study [11].

Regarding the validation of our tissue classification model, the two-fold cross-validation with random sampling was performed spectrum-wise and not patient-wise. Hence, the data from different patients can be present in both training and test sets, but the same spectrum will only belong to either the training set or test set for each fold of cross-validation. However, this does not mean that the results of this study are invalid or that classification performance metrics have been overestimated. It is important to consider that the intra-patient is comparable to the inter-patient variation in our dataset (the standard deviation of measurements within each individual patient is less than twice the standard deviation of all measurements of all patients). A typical mean and standard deviation across all measured locations of one patient for both tumor and normal mucosa measured with both probes is given in Appendix A. No clear trend has been identified in the data of specific patients, possibly due to the limited number of patients. If such a trend is identified in future studies, a patient-wise validation of our model would estimate the classification performance metrics of Table 2 and Table 3 most likely as robustly. Since this trend has not been identified, a spectrum-wise validation of our tissue classification model could potentially be stronger because this validation includes the data of more patients compared with a patient-wise validation. This is especially important given that our study was limited to 47 patients.

Furthermore, the ultimate aim of our study is to classify normal mucosa and cancer tissues at each location within the same patient. Therefore, robust validation of our tissue classification model should consider both measurements of different patients as well as measurements from the same patient, but at different tissue locations, as we did in our study.

Our study focused on biochemical and DRS spectral differences between mucosa and cancer tissues, which, from a clinical perspective, would be especially useful for guidance on cancer margins and tumor delineation. Future studies exploring tumor detection will include data from tissues with non-cancer pathology. This would include a variety of non-neoplastic processes commonly seen in CRC patients including inflammatory bowel disease, radiation-induced fibrosis, and scarring following local excision of cancers.

From a research perspective, the results of our study can be used to design optimized optical instruments targeting the specific wavelength ranges and tissue probed depth, as well as to evaluate the feasibility of employing biochemical analysis methods targeting Hb, HbO_2_, water and lipid for tissue classification. These methods include other optical sensors and/or combinations with existing instruments for CT, MRI, electrical measurements, mass spectrometry, and others. Simulations of light transport in tissues and DRS measurements may benefit from estimating accurate data of the chromophores most relevant for tissue classification. The same applies to the creation of 3D cell models (e.g., spheroids) mimicking mucosal and cancerous tissues in tissue engineering studies, where the right tissue types and thicknesses should accurately reproduce the tissue biochemistry.

## 5. Conclusions

In the present study, we evaluated the most important tissue biomarkers and wavelength ranges for CRC detection by using diffuse reflectance spectroscopy in the visible/near-infrared wavelength range from 350 nm to 1920 nm. In this range, the biomolecules most relevant for classification between normal and cancer tissues were Hb, HbO_2_, lipid and water. Tissue classification using Hb and HbO_2_ data can be achieved with wavelength ranges 350–590 nm or 600–1230 nm for superficial tissue (short-SDD probe), and 380–400 nm, 420–610 nm, and 650–950 nm for deeper tissue layers (long-SDD probe). Information collected in those wavelength ranges is redundant and the combination between them did not enhance the classification. On the other hand, water and lipid information are complementary and may improve cancer detection and investigation of carcinogenesis upon extension of the wavelength range. Our results suggested that information from deeper tissue layers either accessed by probing the optical window or through the long-SDD probe can differentiate normal and cancer tissues more accurately. Wavelength ranges and probe geometrical configuration used in this study may be used for more specific future instrument design. From a practical perspective, these wavelength ranges and geometrical configurations will be used to develop an optical system for CRC detection during colonoscopy and intra-operatively. By optimizing the features for discrimination between normal and cancerous tissues, tissue identification is performed in real-time with a single reading of about 2–3 s. By means of the integration of real-time tissue identification into a flexible fiberoptic probe which could be passed down the working channel of an endoscope, optical spectroscopy may provide a powerful tool that can be used to detect cancer cells and direct management in real time. In addition, this spectroscopic technique can detect more subtle mucosal abnormalities such as sessile serrated polyps, which may be difficult to identify during colonoscopy. Once instruments with wavelength ranges and probe geometrical configuration found in this study have been miniaturized and integrated into colonoscopies, next-generation instruments can be manufactured and their impact in reducing CRC morbidity and morbidity can be assessed in future *in vivo* studies.

## Figures and Tables

**Figure 1 cancers-14-05715-f001:**
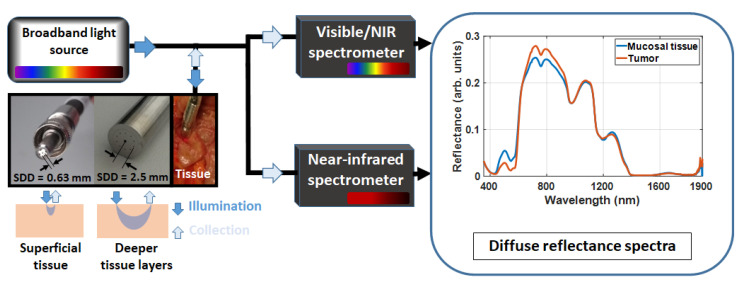
Schematic drawing of our DRS system. The obtained broadband reflectance contains information about a larger variety of tissue biomolecules compared to studies probing shorter wavelength ranges [61,86,87,102,103,104,105,106,107,108,109,110,111]. Broadband reflectance spectra were obtained by merging the visible and NIR spectra based on the overlapping spectral region between the two spectrometers (from 1090 nm to 1140 nm). The spectral merging procedure is described in detail in [11,12] and in Section 2.5. Briefly, in the overlapping spectral region, the two spectra are added with a smoothing weighting.

**Figure 2 cancers-14-05715-f002:**
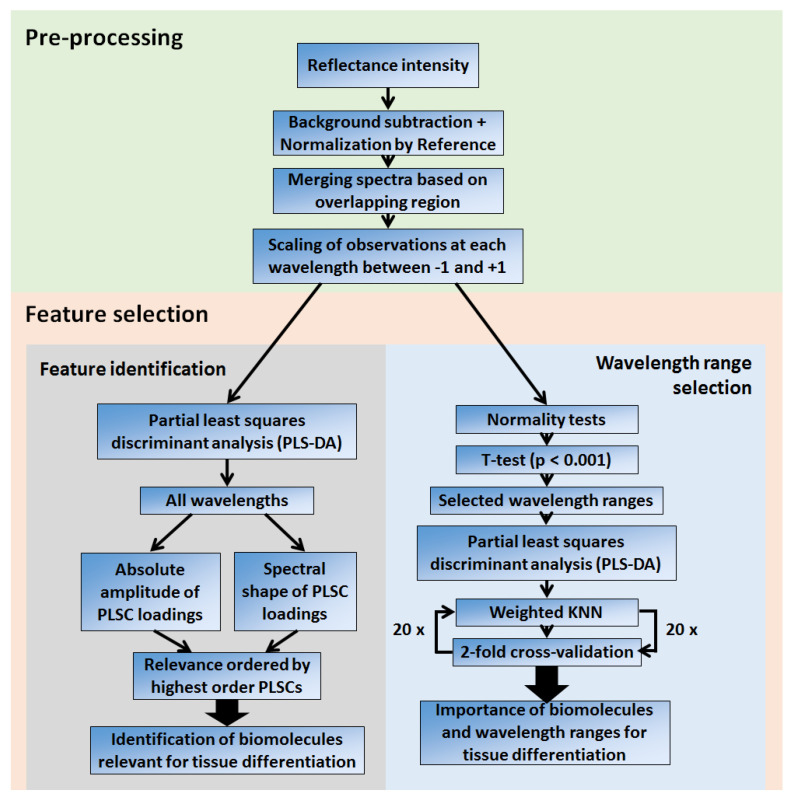
Flowchart of the steps of our spectral analysis.

**Figure 3 cancers-14-05715-f003:**
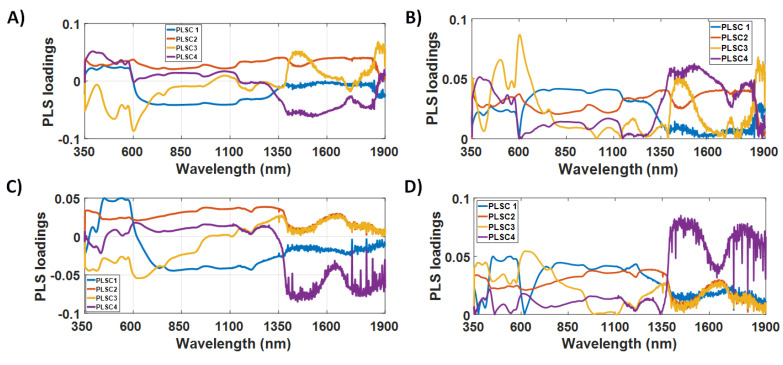
PLS components (PLSCs) for the classification between cancerous and mucosal/submucosal tissues. (**A**) Raw and (**B**) absolute values of PLSC loadings for the short-SDD probe and (**C**) Raw and (**D**) absolute values of PLSC loadings for the long-SDD probe.

**Figure 4 cancers-14-05715-f004:**
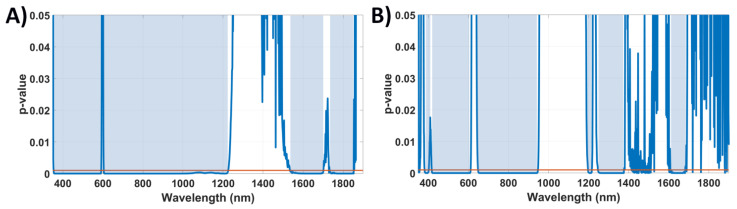
Selected spectral regions (blue) where statistically significant differences (*p* < 0.001) are found for (**A**) short-SDD probe and (**B**) long-SDD probe. The red line indicates the 0.001 cutoff for the *p*-value.

**Figure 5 cancers-14-05715-f005:**
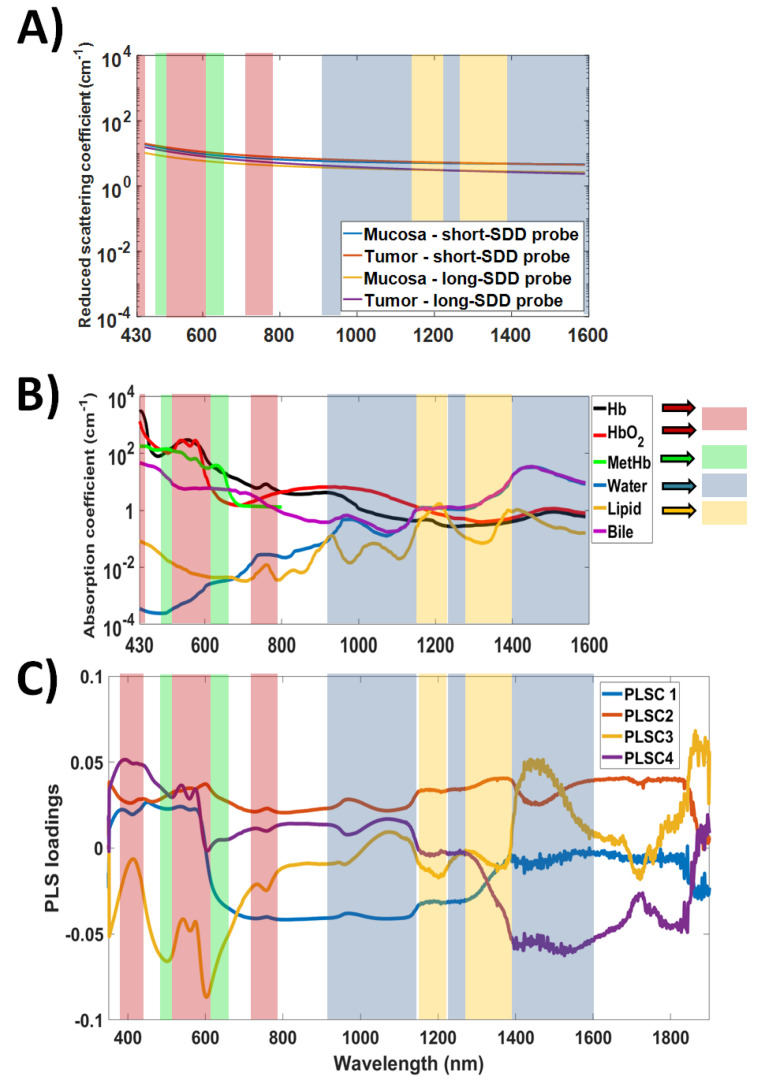
Wavelength regions with spectral features of mucosa and tumor scattering coefficients and each tissue chromophore shown at the (**A**) Reduced scattering spectra of mucosa and tumor tissues of both short-SDD and long-SDD probes, (**B**) chromophore absorption spectra and (**C**) PLSCs of the short-SDD probe.

**Table 1 cancers-14-05715-t001:** Patient demographics, cancer types and tumor staging classification.

Patient and Cancer Characteristics		Number of Patients/Tumors
**Total**		47
**Gender**	Male	32
	Female	15
**Age (years)**	Median	69
	Minimum	40
	Maximum	89
	Interquartile range	13.5
**Cancer types**	Adenocarcinoma	47
**T (tumor) stage**	pT1	5
	pT2	7
	pT3	26
	pT4	9
**N (lymph node) stage**	N0	19
	N1a	9
	N1b	12
	N1c	1
	N2	1
	N2a	4
	N2b	1

**Table 2 cancers-14-05715-t002:** Tissue classification performance * of PLS-KNN for the short-SDD probe using wavelengths selected by *t*-test (*p* < 0.001). Blue fields represent the performance using visible/NIR light detection available in Si-detector-based spectrometers, orange fields cover the performance using NIR wavelengths detected by InGaAs-based spectrometers and green fields show the performance of both types of wavelength range combined. Means and standard deviations were taken from the outcomes of 20 iterations of 2-fold cross-validation.

Wavelengths	Sensitivity	Specificity	Accuracy	AUC
**350–540 nm, 540–590 nm**	(78.2 ± 0.9)%	(75.8 ± 1.6)%	(77.1 ± 1.0)%	(0.854 ± 0.007)
**350–590 nm**	(78.4 ± 1.1)%	(74.9 ± 1.4)%	(76.7 ± 0.8)%	(0.85 ± 0.005)
**600–1230 nm**	(79.8 ± 0.9)%	(84.4 ± 1.4)%	(81.9 ± 0.8)%	(0.894 ± 0.005)
**350–590 nm, 600–1230 nm**	(78.6 ± 0.7)%	(75.4 ± 1.0)%	(77.1 ± 0.7)%	(0.854 ± 0.006)
**1530–1700 nm**	(70.9 ± 1.1)%	(67.0 ± 1.6)%	(69.1 ± 1.0)%	(0.771 ± 0.009)
**1730–1850 nm**	(69.1 ± 1.0)%	(69.5 ± 1.3)%	(69.3 ± 0.9)%	(0.765 ± 0.007)
**1530–1700 nm, 1730–1850 nm**	(76.4 ± 0.9)%	(77.7 ± 1.1)%	(77.0 ± 0.7)%	(0.845 ± 0.006)
**350–590 nm, 600–1230 nm,** **1530–1700 nm, 1730–1850 nm**	(85.5 ± 0.8)%	(84.0 ± 1.0)%	(84.8 ± 0.7)%	(0.919 ± 0.004)
**350–1920 nm**	(85.6 ± 0.9)%	(80.4 ± 1.1)%	(83.2 ± 0.8)%	(0.905 ± 0.005)

* Results corresponding to 2-fold cross-validation from 20 iterations of random sampling of training and test sets. Our validation provides a sufficiently robust estimation of classification performance metrics since the standard deviation of measurements within each individual patient is less than twice the standard deviation of all measurements of all patients. In addition, no clear trend among specific patients has been identified in our dataset of 47 patients. A typical measurement means and standard deviation for one patient can be found in Appendix A.

**Table 3 cancers-14-05715-t003:** Tissue classification performance * of PLS-KNN for the long-SDD probe using wavelengths selected by *t*-test (*p* < 0.001). Blue fields represent the performance using visible/NIR light detection available in Si-detector-based spectrometers, orange fields cover the performance using NIR wavelengths detected by InGaAs-based spectrometers and green fields show the performance of both types of wavelength range combined. Means and standard deviations were taken from the outcomes of 20 iterations of 2-fold cross-validation.

Wavelengths	Sensitivity	Specificity	Accuracy	AUC
**380–400 nm**	(86.0 ± 0.9)%	(85.0 ± 0.9)%	(85.6 ± 0.7)%	(0.925 ± 0.004)
**420–610 nm**	(85.6 ± 0.5)%	(87.2 ± 0.6)%	(86.3 ± 0.3)%	(0.93 ± 0.004)
**650–950 nm**	(89.6 ± 0.6)%	(89.7 ± 1.0)%	(89.7 ± 0.6)%	(0.96 ± 0.004)
**380–400 nm, 420–610 nm,** **650–950 nm**	(87.0 ± 0.8)%	(85.5 ± 0.8)%	(86.3 ± 0.7)%	(0.931 ± 0.003)
**1200–1220 nm**	(67.8 ± 1.2)%	(63.3 ± 1.9)%	(65.7 ± 1.1)%	(0.707 ± 0.013)
**1250–1380 nm**	(77.1 ± 1.0)%	(80.7 ± 1.0)%	(78.8 ± 0.7)%	(0.87 ± 0.006)
**1600–1690 nm**	(62.4 ± 1.1)%	(58.3 ± 1.6)%	(60.4 ± 0.8)%	(0.654 ± 0.008)
**1200–1220 nm, 1250–1380 nm, 1600–1690 nm**	(77.6 ± 1.0)%	(84.7 ± 1.1)%	(81.0 ± 0.8)%	(0.883 ± 0.006)
**380–400 nm, 420–610 nm,** **650–950 nm, 1200–1220 nm, 1250–1380 nm, 1600–1690 nm**	(89.1 ± 0.7)%	(90.2 ± 0.7)%	(89.6 ± 0.5)%	(0.957 ± 0.004)
**350–1920 nm**	(89.3 ± 0.6)%	(90.2 ± 0.7)%	(89.7 ± 0.5)%	(0.959 ± 0.003)

* Results corresponding to 2-fold cross-validation from 20 iterations of random sampling of training and test sets. Our validation provides a sufficiently robust estimation of classification performance metrics since the standard deviation of measurements within each individual patient is less than twice the standard deviation of all measurements of all patients. In addition, no clear trend among specific patients has been identified in our dataset of 47 patients. A typical measurement means and standard deviation for one patient can be found in Appendix A.

**Table 4 cancers-14-05715-t004:** Scattering and absorption features of the superficial tissue PLSCs.

			Scattering and Absorption Features					
**PLS (Short-SDD probe)**		**VIS Scat**	**NIR Scat**	**Hb**	**HbO_2_**	**MetHb**	**Water**	**Lipid**
	**PLSC1**			X	X		X	
	**PLSC2**	X	X	X	X		X	
	**PLSC3**			X	X		X	X
	**PLSC4**			X	X		X	X
**PLS (Long-SDD probe)**	**PLSC1**			X	X		X	X
	**PLSC2**	X	X	X	X		X	X
	**PLSC3**			X	X		X	X
	**PLSC4**			X	X		X	X

## Data Availability

The data presented in this study are available on request from the corresponding author.

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
