# Peer review of "Insights into Biochemical Sources and Diffuse Reflectance Spectral Features for Colorectal Cancer Detection and Localization"

_cancers, 2022, doi:10.3390/cancers14225715_

Round 1

Reviewer 1 Report

In this paper, Nogueira and co-authors perform an extensive study on surgically resected colon specimens, taking diffuse reflectance spectra at two source-detector separations from cancerous and noncancerous regions of the colon, with probe placement directed by trained surgeons. The data were then analyzed to determine the most diagnostic wavelength ranges and the relative contributions of individual biomolecules to the diagnostic process. Overall, the study is detailed and well-written, and I would recommend its publication in Cancers following minor revisions. There are a few points I think deserve a bit more discussion and clarity for the reader, as discussed below:

1. On line 247 the authors scale the intensities to range between -1 and +1 (presumably after mean centering?). Why not scale the variances - i.e. operate on the Z-scores?

2. The PLSC training was not clear. Were the PLSCs calculated on the entire dataset (i.e. no separate training/validation or cross-validation scheme)? If so, this should be clearly mentioned in the text, since it means the achieved accuracies are not very relevant for prospective studies (although the discussion of the determine loading shapes is still valid!).

3. Showing the distribution of the spectra in the PLS loading space could also be highly informative, to show how important each PLS dimension is to the classification process. In other words, if the data are more-or-less perfectly separated on the PLS1 axis, then the importance of PLS2, 3, etc. is less relevant.

4. Related to the cross validation of the KNN scheme, was the 2-fold cross validation performed on the spectrum level or sample level? I.e. did spectra from the same patient participate in both the test and validation sets simultaneously? Obviously segregating the test and validation sets on the patient-level would be more robust, but possibly more limited given the limited patient number. Clarifying this would help the reader put the results into the appropriate context.

5. The discussion of the biomolecule contribution to each PLS loading was mostly qualitative. It would have been more convincing to see quantitative fits of pure biomolecule spectra to the PLS loadings. This is especially true because the authors claim that scattering was only observed strongly in PLS2, but it may have been present as a more "invisible" polynomial offset that would only have been revealed by some kind of spectral fitting process.

6. Related to the role of scattering in this study, the source-detector separations were both quite short, such that the photon paths probed by the short-path detector were probably "subdiffusive" rather than diffuse photons. If the source-detector separation was chosen to be longer (which may not be viable due to the desire to fit everything into an endoscope), wouldn't one expect the scattering to become more relevant? Or, to turn it around, isn't the reason scattering is not relevant in this dataset due to the detection geometry, rather than something fundamental about cancer-related alterations to tissue scattering? This seems to deserve some discussion in the manuscript.

7. VERY minor grammar point that stuck out to me: twice in the manuscript the phrase "keeping the tissue moisture" was used, which should be corrected to "keeping the tissue moist".

To conclude, the authors present a detailed study of the major wavelength regions and biomolecule contributors to diagnosis of colorectal cancer in an endoscopic geometry. Such a study should be of interest to the readership of Cancers, and should help future prospective studies using reflectance spectra in endscopic cancer diagnostics. However, several points related to the sample processing and interpretation of the results should be addressed/clarified prior to publication.

Reviewer 2 Report

The study is interesting, but there are major issues with classifier design and validation (see pts. 4-6) which should be addressed first.  

1)      Introduction - lines 75-92 need more literature refences.

2)      Figure.1. Indicate wavelength(s) or their range where the spectra are concatenated. How was the spectra merging performed? The figure does not show any difference in reflectance units at the place where the spectra were merged. Please describe these effects (if there are any) and how you dealt with them.

3)      Describe the process of centering the long and short-SDD probes exactly in the same place of rumor and normal, tissues for spectra merging and collection. Otherwise, explain why probe centering is not important in order to preserve the uniqueness of from tumor and normal mucosa.

4)      The selection of spectral bands using a statistical test followed by subsequent tissue classifications with the same pre-selected bands seems to be flawed from the machine learning standpoint. The problem is that spectral bands were selected using all available datapoints (spectral profiles from all tissues and all regions) and then the same cases were used for classification. A more appropriate approach is to use a subset of cases (discovery set) to identify differential spectral bands (p-value<0.05) then perform PLSC to engineer new features and subsequently train a classifier. The classifier should be then applied to the other set of cases (validation or test set) to observe its performance.

5)      Random selection of spectra from cancer and normal areas regardless the case for training will likely lead to poor generalization of the classifier and biased classifier accuracy.  The training and test sets should contain data from unique cases, that is, all spectra from a case should be placed either in the training or test sets.  If the case-wise formation of training and test sets is not ascertained, the knn classifier may choose spectra from the same case (acquired from a different spot) during classification and hance yield overoptimistic performances.  I recommend that authors look up the literature on classifier validation and training/test set preparations. Here is a useful example https://web.stanford.edu/~hastie/ElemStatLearn/printings/ESLII_print12.pdf (pg. 241 and section 7.10.2).

6)      A figure showing variability of recorded spectra (both tumor and normal, both probes) across all 15 points within a case would be beneficial for the readers. The authors should discuss robustness of collected spectra for the classification of tissues and how the variability of spectra across cases can affect classification performance.

7)      As the plots in Figure 3 are helpful in understating plots in Figure 4, consider presenting figure 3 before figure 4.

8)      Figure 5. There is likely a typo in the figure legend: “absorption spectra 6”.

Reviewer 3 Report

The paper by Nogueira et al describes application of diffuse reflectance spectroscopy for the detection of colorectal cancer. The theme of the study is important and interesting as this study provides analysis  of biochemical components of tissues that contributes to diffuse reflectance spectra. The paper is well written and have quite full explanations in the Introduction, Materials and Methods, Results and Discussion. The number of analyzed tissues is not large, but at the same time such number of samples allows for aquiring of relible results.
In my opinion, the experimental part of the study is free of drawbacks, but at the same time there are some major issues regarding statistical analysis of spectral data:

1. The authors utilized in the analysis 4 Principal components (PCs), however, the correct determination of adequate PCs number is quite complicated. May the authors explain how the number of PCs sutable for the analysis was determined? Note here that shape of utilized PCs looks adequate for the analysis, but maybe it is possible to utilize some PCs of higher order? The authors provides Refs 64 and 65 to explain how PLS is performed, but exact explanation on the analyzed spectral dataset is required to prove that the authors utilize correct number of PCs.

2. The most crucial question is regarded to the application of 2-fold cross-valdiation (CV). It is very fine that the authors utilized CV to check the stability of the constructed models, but at the same time during such CV it is very likely that the spectra from one tissue sample will appear both in training and in validation set. The authors analyzed only 47 patients but collected more than 1000 spectra. Thus, in random CV the spectra from one tissue sample will be utilized in the model to construct the classifier, and at the same time almost the same spectra (as variation of spectra for one sample is commonly statistically insignificant) will appear in validation set. In this regard, the model already knows what kind of spectra will appear in validation set, and it is possible that high performance of classification model is due to incorrect spectral data splitting. If I understand correctly the authors utilized random cross-validation also in their previous studies (Ref 70) and some deeper analysis is required to prove that the proposed classification models are correct. It is advised to utilize k-sample-out CV in which all spectral data from one tissue sample is presented only in one set (in validation or in training).

Some additional explanations regarded to common drawbacks in spectral data analysis may be found in recent publication [doi.org/10.48550/arXiv.2210.10051] (and the authors may aknowledge this publication if they found it useful). 

Some minor issues:

1. Very commonly loadings in PLS or in other analysis as PCA are called principal components (PCs) but not PLSCs (partial least squares components). Maybe it is better to call loadings as PCs  for better understanding of the readers?

2. Ref 69 dedicalted to the analysis of leafs, maybe it is better to replace this reference with reference dedicated to the analysis of human tissues?

In general, the paper may be published after elimination of arised issues. Detailed explanations from the authors are required to prove the obtained classification models stability.

Round 2

Reviewer 2 Report

The authors addressed my comments. Regarding the classifier design, I still think that case-wise selection of spectra for training and testing would be more appropriate.  An more in-clinic like testing scenario would be to apply the leave-one-case out cross validation (LOOCV) instead of the 2 fold cross validation.  The LOOCV would give authors mode training data and would permit testing the classifier on single cases.  However, give the early stage of this research project the 2-fold cross validation would be acceptable for this report.  

Reviewer 3 Report

The authors addressed arised issues.

The paper may be published.